# Enhancement of Marine Lantern’s Visibility under High Haze Using AI Camera and Sensor-Based Control System

**DOI:** 10.3390/mi14020342

**Published:** 2023-01-29

**Authors:** Jehong An, Kwonwook Son, Kwanghyun Jung, Sangyoo Kim, Yoonchul Lee, Sangbin Song, Jaeyoung Joo

**Affiliations:** 1Lighting & Energy Research Division, Korea Photonics Technology Institute 9, 500-460 Cheomdan Venture-ro 108beon-gil, Buk-gu, Gwangju 61007, Republic of Korea; 2Department of Electrical Engineering, Yeungnam University, Gyeongsan 42415, Republic of Korea

**Keywords:** computer vision, embedded system, deep learning, dehaze, LED marine lantern, serial communication, artificial intelligence, sea fog

## Abstract

This thesis describes research to prevent maritime safety accidents by notifying navigational signs when sea fog and haze occur in the marine environment. Artificial intelligence, a camera sensor, an embedded board, and an LED marine lantern were used to conduct the research. A deep learning-based dehaze model was learned by collecting real marine environment and open haze image data sets. By applying this learned model to the original hazy images, we obtained clear dehaze images. Comparing those two images, the concentration level of sea fog was derived into the PSNR and SSIM values. The brightness of the marine lantern was controlled through serial communication with the derived PSNR and SSIM values in a realized sea fog environment. As a result, it was possible to autonomously control the brightness of the marine lantern according to the concentration of sea fog, unlike the current marine lanterns, which adjust their brightness manually. This novel-developed lantern can efficiently utilize power consumption while enhancing its visibility. This method can be used for other fog concentration estimation systems at the embedded board level, so that applicable for local weather expectations, UAM navigation, and autonomous driving for marine ships.

## 1. Introduction

A navigational aid is a device that prevents marine accidents and supports the safe navigation of ships. Among various devices, the marine lantern, which guides ships in the sea, is a lighting device that displays the position of the route marker.

According to statistics from the Korea Maritime Police Agency in 2017, yellow dust, fine dust, rain, snow, and sea haze caused maritime safety accidents. Among them, haze (sea fog) lowers the visibility of marine lanterns as a significant factor in accidents. Over the past five years, the number of ship accidents and casualties due to sea fog in Korea was 3586 and 407 ships, respectively, and the number of accidents is increasing.

Currently, sea fog is coped with using sonar that sounds like a horn as a temporary measure in a bad visibility situation. However, its use is decreasing due to complaints such as noise generation. Commercial marine lanterns turn on/off depending on day or night, and are thus effective as a navigation aid up to a specific concentration of sea mist and prevent marine safety accidents. However, under heavy sea fog, the luminous distance of unmanned lighthouses or breakwater lighthouses drastically decreases, which poses a significant risk to maritime traffic safety. Therefore, it is essential to install a navigation system that can ensure visibility through the sensor, even in sea fog conditions. In addition, the usage of lanterns does not control automatic light intensity using wireless communication according to visual range by a short-distance marine lantern or IR remote control light intensity adjustment because of marine environmental conditions.

Current marine lanterns consume limited battery power and illuminate the same luminous intensity, regardless of visual range, i.e., inefficient power loss on a bright day or low visibility on foggy days.

For example, middle power-consuming marine lanterns have power consumption (80 W)—floating light intensity (3500 cd), high power (132 W, 46,700 cd), and low power (50 W, 4300 cd).

This research proposes the marine lantern horizontal light distribution technology using a single LED light source with a double reflector structure, which consumes relatively large battery power compared to the direct LED emitting lantern with some dark zone but provides fulfilled luminous intensity distribution, i.e., no dark zone. 

This paper used an NVIDIA Jetson Nano embedded board, which was installed and executed in a natural marine environment, a waterproof and dustproof camera sensor, and an LED marine lantern [1,2,3,4]. The sea fog image was taken with the camera sensor, and the sea fog removal image was obtained using the deep learning (CNN)-based dehaze technique (FFA-Net [5], Dehazenet [6,7,8,9,10,11,12,13,14,15]) on the embedded board. To determine the sea fog concentration, the PSNR (peak signal-to-noise ratio) and SSIM (structural similarity index map) [16] were applied to the images before and after sea fog removal to derive a numerical value. The luminous intensity was designated according to the derived value, i.e., haze level and increased marine lantern’s intensity through serial communication. Finally, an intelligent intensified control system was demonstrated in a test bed, in which the brightness of the marine lantern was autonomously controlled according to the concentration of sea fog.

By using the developed algorithm and embedded system, we have developed a novel marine lantern with a double reflector structure with high floating light intensity (4500 cd) and relatively low power consumption (72 W). 

We validated the automatic visibility enhancement system of the developed lantern at the Korea Institute of Construction Technology’s Meteorological Re-realization Center (KICTMREC). The concentration of sea fog haze level was artificially controlled to confirm the visibility distance and brightness control according to the concentration of sea mist level.

## 2. Materials and Methods

For this development, we used image data collection and selected a proper embedded board with cameras. The optimum algorithm training and testing were conducted. In addition, an LED marine lantern was used in an outdoor test bed environment.

### 2.1. Embedded Board/Camera Comparison and Selection

For this remote application in the coastal area, embedded boards for IoT are more suitable than desktops for less power consumption and small installation volume. After comparing their specifications, three types of embedded boards were recommended. Regarding the implementation environment, Latte Panda Alpha has a compatibility problem when applying a deep learning model based on Windows OS. Also, Raspberry Pi 4 can implement deep learning models based on Linux OS, but the computation speed is slow. The Jetson Nano board is compatible with implementing deep learning models based on Linux OS, and it is faster than other embedded boards by CPU and GPU operation. In addition, its average power consumption is about 10 W~15 W. Therefore, the Jetson Nano was finally selected as a proper embedded board. Table 1 below shows a comparison of the embedded board specifications.

A commercially available waterproof camera sensor with a relatively low price was preferred, which is appropriate for commercial marine applications. The ELP USB webcam ($90 US) was compatible with the NVIDIA Jetson Nano embedded board and operated well under waterproof conditions. Figure 1 below shows the embedded board, and ELP camera used.

### 2.2. Data Collection and Construction

For artificial intelligence (deep learning) model training and testing, haze image and standard clear image data were collected. We also captured and used actual sea fog imaging and sea fog test bed imaging. 

The Reside dataset [17], O-Haze dataset [18], and Dense dataset [19] were collected for the open image data set. The actual sea fog images were collected from Pukyong National University (PNU) in Busan, South Korea, toward the coast. The sea fog test bed images were collected after arbitrarily creating a sea fog environment at the KICTMREC. In addition, customized data were constructed to increase the utilization of the collected images by adjusting the sharpness and brightness of the sea fog image data. Figure 2 below shows an example data set.

In the case of actual sea fog imaging data collection, photos were taken in the coastal direction at the PNU. Image data from the captured video were utilized for AI learning. Figure 3 below shows the result of the actual sea fog image collection.

The captured images in the sea fog test bed at KICTMREC were filmed according to the sea fog concentration and utilized for AI learning. Figure 4 below shows the results of images collected from the sea fog test bed, and Figure 5 shows an example of image sharpness_brightness correction within the collected data set.

### 2.3. Image Dehaze Technique and Sea Fog Concentration Value Derivation

#### 2.3.1. Image Dehaze Technique

FFA-Net is a method that separates channels and pixels and then converges them to increase flexibility and expressive performance. It was implemented in the GPU server environment and used the NVIDIA RTX2080Ti graphics card for its higher performance among the state-of-the-art (SOTA) models in the outdoor haze data set. Moreover, it showed excellent haze removal performance according to the haze distribution in the image. 

Dehazenet is based on convolutional neural networks (CNN), and layers of Maxout units are used for feature extraction, which can generate almost all haze-relevant features. A novel nonlinear activation function in Dehazenet, called the Bilateral Rectified Linear Unit (BReLU), is proposed. This technique can improve the image quality from which haze (sea fog) has been removed and reduces the learning time of the CNN architecture by utilizing the image patch applied with the transmission label. It was used in the NVIDIA Jetson Nano embedded board. Figure 6 below shows the structure diagram of Dehazenet.

For the basic algorithm for this research, the SOTA model was selected for the implementation environment and compatibility and some code editability. There are many different methodologies to remove the hazing effect and achieve clear original images. However, we have focused on finding the level of haze more accurately with captured original images by applying the dehaze algorithm.

In recent research, several dehaze algorithms were trained and tested from various sea or lake foggy image data sets, such as an end-to-end sea fog removal network using a multiple scattering model (AEESFRN) [20]. From this reference, the DCP [21,22,23,24,25] and GCAN [26] models have been applied to the embedded boards based on image processing, but their dehaze performance is hard to apply for accurate control of dehaze concentration. In the case of CAP [27], AOD-Net [28], deep learning is used, and performance is reliable, but the model size is too large, so the computation speed is slow. 

By comparing their dehazing performance and finding accurate haze control parameters in Table 2, Dehazenet with a deep learning process, was conducted for better accuracy and performance with any implementation errors on the NVIDIA Jetson Nano Board. All libraries and programs that we have developed was compatible with the deep learning process. In addition, the developed system showed fast operation speed and intuitive data learning and code modification. For another algorithm suitable for your GPU server environment, FFA-Net was also verified for better dehaze models for comparison.

We have utilized open data sets with captured on-site sea fog images for test and training. First, we collected the proper outdoor haze image from the open data set (Reside dataset, O-Haze dataset, Dense dataset) for the Dehazenet model training process. These images about 25,000 were separately trained in the Dehazenet model and enhanced its original images after dehazing. Those outdoor images did not include real site images according to hazy weather conditions. Therefore, we have captured the on-site marine environment images near PNU and KICTMREC. To magnify the number of data sets, we reproduced about 25,000 image research by adjusting the size, sharpness, brightness, etc.

After confirming the results through learning, those images were retrained on the previously trained Dehazenet model to improve performance. Those image data constructed from the selected model were extracted, learned, and implemented by modifying the convolution layer and hyperparameters. The operating system was Linux, and various related software such as Caffe, NumPy, OpenCV, CUDA, and python was used. After the whole data processing, we finally achieved a proper deep-learning model to find accurate sea fog concentration levels.

#### 2.3.2. Sea Fog Concentration Value Derivation

To define sea fog concentration value, we need to mathematically measure image quality using the PSNR and the SSIM. The PSNR represents image clarity as the sea fog level changes. Therefore, the PSNR directly represents sea fog concentration. In addition, the SSIM was also applied to improve its accuracy, which shows the level of similarity between a selected clear image with that in a sea fog environment.

To determine the sea fog concentration, a numerical value was derived by comparing the original sea fog images with the images from which the sea fog was removed. PSNR and SSIM, which are image quality measurement methods, were used for numerical derivation. The PSNR represents the power of noise for the maximum power a signal can have and is mainly used when evaluating image quality loss information. Equation (1) shows the calculation formula of PSNR.
(1)$$\begin{array}{r} \mathrm{PSNR}=10\log_{10}\left(\frac{{\mathrm{MAX}}_{I}^{2}}{\mathrm{MSE}}\right)\\ =20\log_{10}\left(\frac{{\mathrm{MAX}}_{I}}{\sqrt{\mathrm{MSE}}}\right)\\ =20\log_{10}\left({\mathrm{MAX}}_{I}\right)-10\log_{10}\left(\mathrm{MSE}\right) \end{array}$$

${\mathrm{MAX}}_{I}$ is the maximum value of the image, and in the case of an 8-bit grayscale image, it becomes 255. (2) is the mean square error of MSE.
(2)$$\mathrm{MSE}=\frac{1}{mn}\overset{m-1}{\underset{i=0}{\sum }}\overset{n-1}{\underset{j=0}{\sum }}{\left[I\left(i,j\right)-K\left(i,\;j\right)\right]}^{2}$$

*I* is a grayscale image of size (*m* × *n*), and *K* is an image with noise in *I*, i.e., a distorted image. Since there is MSE in the denominator in (1), the smaller the MSE, the larger the PSNR. Thus, a good-quality image will have a relatively large PSNR, and a poor-quality image will have a relatively small PSNR.

PSNR is a suitable method for evaluating quality loss information. However, it often yields quality figures that do not match what people feel because it evaluates image quality by the numerical difference between the original and distorted images. For example, both images have similar PSNR values, but the perceived quality is different.

To overcome this limitation of PSNR, an SSIM was also used. SSIM is a method for evaluating human visual quality differences, not numerical errors, and is evaluated through luminance, contrast, and structure. The overall formula is (3), and each functional formula is *l* (*x*, *y*) for luminance, *c* (*x*, *y*) for contrast, and *s* (*x*, *y*) for structure, as shown in Equation (4).
(3)$$\mathrm{SSIM}\left(x,y\right)={\left[l\left(x,y\right)\right]}^{\alpha }\cdot {\left[c\left(x,y\right)\right]}^{\beta }\cdot {\left[s\left(x,y\right)\right]}^{\gamma }$$
(4)$$l\left(x,y\right)=\frac{2\mu_{x}\mu_{y}+C_{1}}{\mu_{x}^{2}+\mu_{y}^{2}+C_{1}}\;c\left(x,y\right)=\frac{2\sigma_{x}\sigma_{y}+C_{2}}{\sigma_{x}^{2}+\sigma_{y}^{2}+C_{2}}\;s\left(x,y\right)=\frac{\sigma_{xy}+C_{3}}{\sigma_{x}\sigma_{y}+C_{3}}\;$$

Assuming that there is an original image x and a distorted image y, SSIM compares the luminance, contrast, and structure of the two images and combines the three items to obtain the correlation coefficient of the images x and y. Accurate sea fog concentration values can be derived and compared using the two methods by an image loss evaluation and similarity measurement.

### 2.4. Prototyping the Measurement Kit and Serial Communication

#### 2.4.1. Prototyping the Measurement Kit

To capture the sea fog environment in an actual marine environment, we designed and prototyped a measurement kit containing a selected board and camera to prevent the signal error from external exposure. After securely fixing the kit, the case-side cable connection cover was added. Figure 7 shows the designed 3D model.

#### 2.4.2. Serial Communication

For low data loss, even with long-distance communication, the serial communication method was used to control the brightness of the marine lantern with the derived sea fog concentration value, as shown in Figure 8. Communication protocol follows the physical standard of RS232, and the serial settings are baudrate: 9600, data: 8 bit, stopbit: 1 bit, parity: N. It communicated using a master–slave structure. The configured protocol consists of ASCII-based data communication. According to the sea mist concentration, 11 control codes were used to control the light intensity by transmitting control packets. Table 3 shows 11 packet structures for controlling the light intensity of marine lanterns.

A virtual serial port was used to check the control commands and packets transmitted from the existing marine lantern control program. A virtual system was implemented to check the sending and receiving of packets on the PC. As a result of receiving data through the virtual system, it was confirmed that 2 bytes of LF and CR were added to the packet END and then added to the embedded board transmission packet. For the CRC checksum, the data from the bytes after the start packet ‘$’ to before the checksum separator ‘*’ are XOR. As a result of converting to ASCII code, the result value was derived through XOR hexadecimal operation. After the marine lantern’s light intensity control was classified into 20 steps from 0 to 100, the packets were tabled and applied to the embedded board, as shown in Figure 9.

### 2.5. Prototyping LED Marine Lantern

For the demonstration, a medium-sized LED marine lantern was prototyped with a double reflection system to verify the thermal degradation of luminous intensity level, which may be able to satisfy the commanding higher current level under heavy sea fog environment. The maximum temperature of the LEDs was simulated at about 97 ℃, which is acceptable and provides stable light illumination of the marine lantern applying a double reflection. Figure 10 below shows the temperature distribution analysis and modeling results of the LED marine lantern.

### 2.6. Haze/Sea Fog Test Bed

For the actual sea environment test, we installed the developed marine lantern and prototyped measurement kit in the KICTMREC, where the artificial haze (sea fog) environment was created and controlled its concentration. The proposed control system was tested under various variables, such as brightness adjustment according to the visibility distance, sea fog concentration, visibility distance measurement, etc. Figure 11 below shows the Korea Institute of Civil Engineering and Building Technology test bed used for data collection and demonstration.

As shown in Figure 11b, fog generators were installed on both sides of the test bed at 10 m intervals in the 200 m section of the tunnel-type shield, and the minimum fog visibility is possible up to 30 m. In addition, it is very suitable for implementing a specific fog concentration (visible distance) and evaluating the light visibility.

## 3. Results

In this research, we built the intelligent marine lantern with an embedded board where our developed SW was operated. Moreover, this system could control the light intensity according to the concentration of sea mist by using a camera and artificial intelligence in an environment. Its operation performance was confirmed in the test bed.

To verify the developed algorithm’s operation and performance, the dehaze algorithm was applied to a proper haze image, as shown in Figure 12. The result of visibly removing sea fog was derived and confirmed relatively sea fog removal performance.

For comparison, we applied our trained Dehaze’s representative algorithms against DCP and FVR [29,30,31,32] with some of marine images to qualify the research results. The reference result showed; PSNR: 10.75, SSIM: 0.83 for FVR and PSNR: 12.37, SSIM: 0.78 for DCP. Our research results showed that PSNR: 14.74, SSIM: 0.89, which can derive better figures from resolution images. Accordingly, the dehaze performance of our algorithms was relatively higher than references, and it was confirmed visually in Figure 13. These results show the comparison of visual images of (a) Original (b) FVR, (c) DCP (d) Our algorithm.

Before installing the system in the test bed, we double-checked its operation performance with the algorithm, as shown in Figure 14. The test was conducted by connecting the camera and the marine lantern. After checking the interlocking operation of each sensor, it was confirmed that the marine lantern light intensity was controlled by arbitrarily inputting the sea mist concentration.

For the outdoor test bed KICTMREC, we compared the performance of our intelligent marine lantern with a commercially available marine lantern (On/Off operation) whose brightness could not change, as shown in Figure 15a. Under designating the distance as 100 m and 200 m, real-time images were captured in 20-min increments to determine the sea fog concentration in real-time images. Based on the derived values of PSNR and SSIM of both images before and after capture, the degree of sea fog concentration was grasped in real time, and the brightness of the marine lantern was controlled through serial communication.

The developed marine lantern system showed superior brightness to the existing marine lantern under heavy sea fog, i.e., poor visibility. The elevation in visual distance of the developed lantern was clearly observed according to the sea fog concentration.

The brightness of the lantern according to the visibility distance was confirmed at 200 m and 100 m distances. As shown in Figure 16a, there is no visual difference if the visibility is good. In Figure 16b–d, it was visually confirmed that the intelligent marine lantern on the left is brighter than the existing marine lantern by autonomous brightness control when the visibility is lowered to 100 m, 50 m, and 10 m.

PSNR and SSIM derived from two images (the image with/without sea fog) were used to control the brightness of the marine lantern according to sea fog concentration.

In order to validate the designed algorithm with the developed embedded system according to the PSNR and SSIM values, we compared the transmitting visibility of the developed system with current static luminous intensity lanterns according to the sea fog concentration value in KICTMREC.

To define the light intensity level along with the PSNR and SSIM values, we divided these values into four levels (0~34,000, 34,000~45,000, 45,000~50,000, and 50,000~120,000) with settings ranging from a minimum of 0 to a maximum of 120,000. By applying these algorithms, the lantern was automatically controlled with four different lights, as the higher the PSNR and SSIM values, the higher the brightness.

By applying the developed AI to the embedded board with a proper camera, we dramatically improved the visual distance of the marine lantern. Thus, the darker the haze, the higher the light intensity of the lantern autonomously.

In addition, this research has many possibilities and meanings for a navigational control system, especially for a marine lantern. Moreover, this can be applied for any light-intensity controlling equipment in a marine environment according to the degree of sea-fog level through an artificial intelligence camera.

## Figures and Tables

**Figure 1 micromachines-14-00342-f001:**
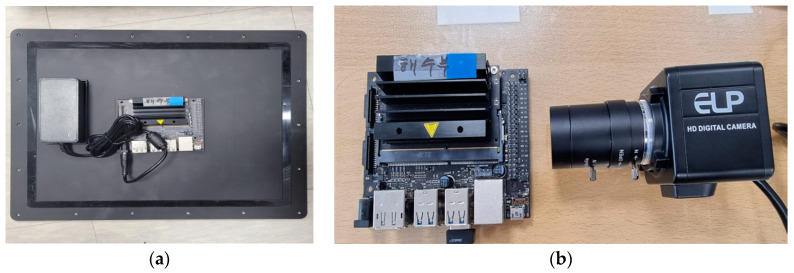
(**a**,**b**) Embedded board and ELP camera module used.

**Figure 2 micromachines-14-00342-f002:**
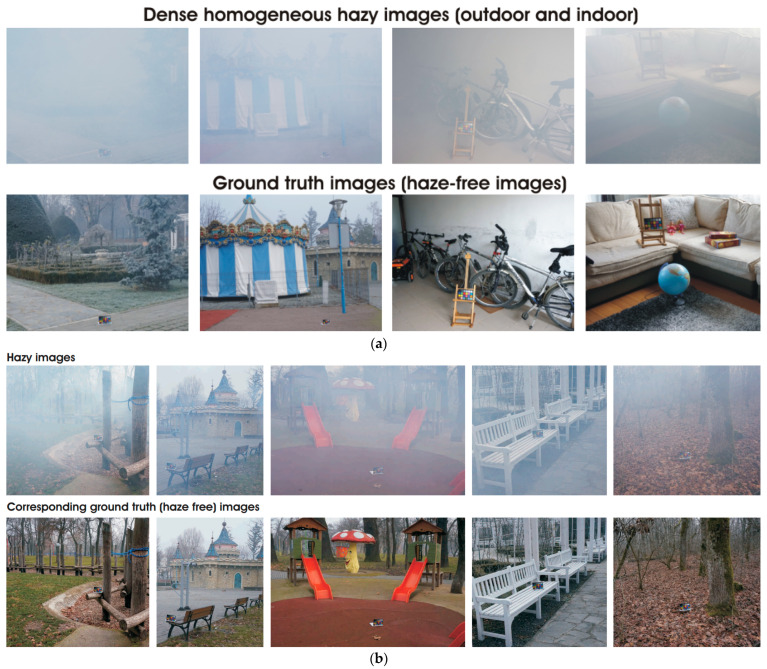
(**a**) Open image data set Dense dataset; (**b**) open image data set O-Haze dataset; (**c**) open image data set Reside dataset.

**Figure 3 micromachines-14-00342-f003:**
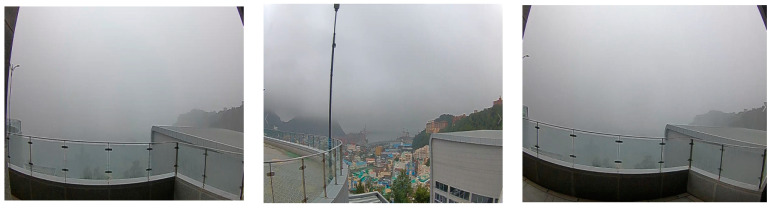
Actual sea fog imaging data collection.

**Figure 4 micromachines-14-00342-f004:**
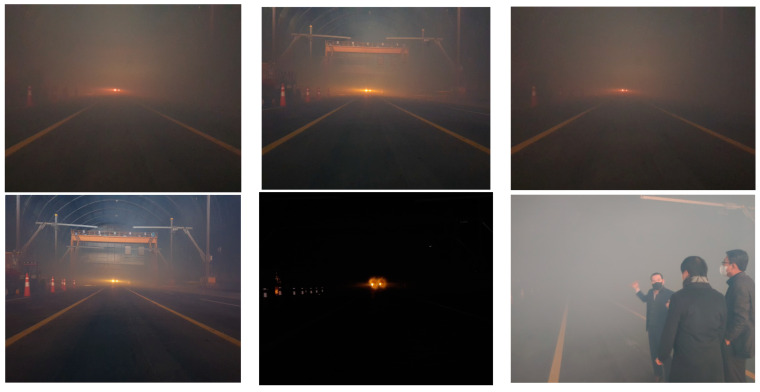
Sea fog test bed imaging data collection.

**Figure 5 micromachines-14-00342-f005:**
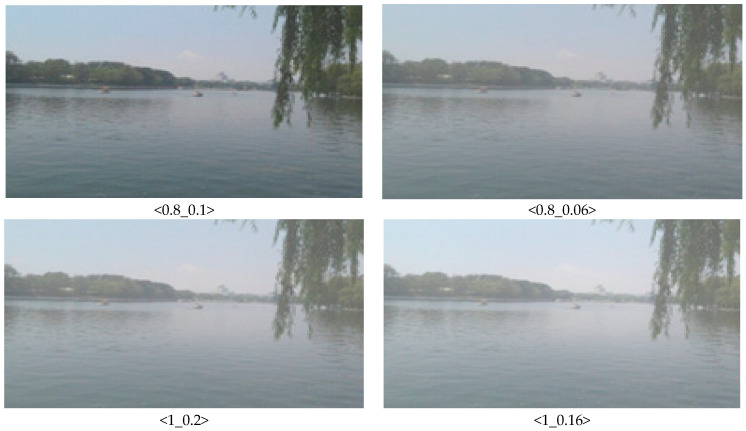
Correction of the collected sea fog image (sharpness_brightness).

**Figure 6 micromachines-14-00342-f006:**
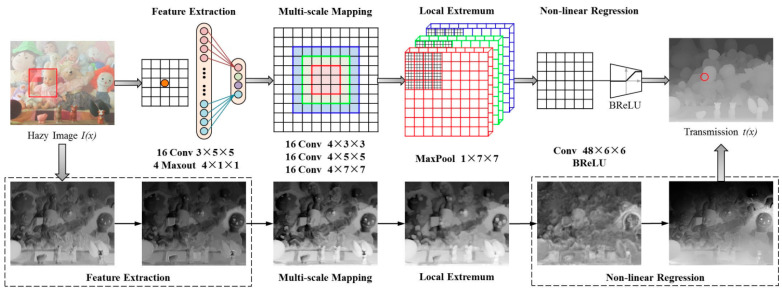
Dehazenet model structure diagram.

**Figure 7 micromachines-14-00342-f007:**
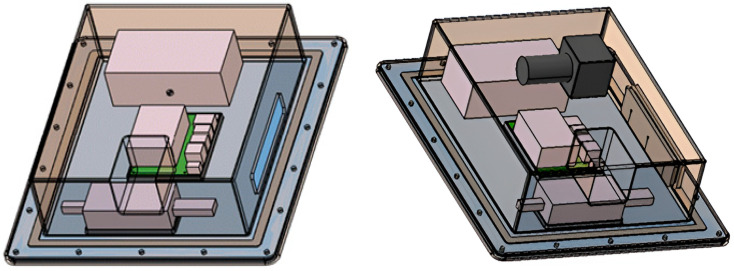
Case modeling using CATIA.

**Figure 8 micromachines-14-00342-f008:**
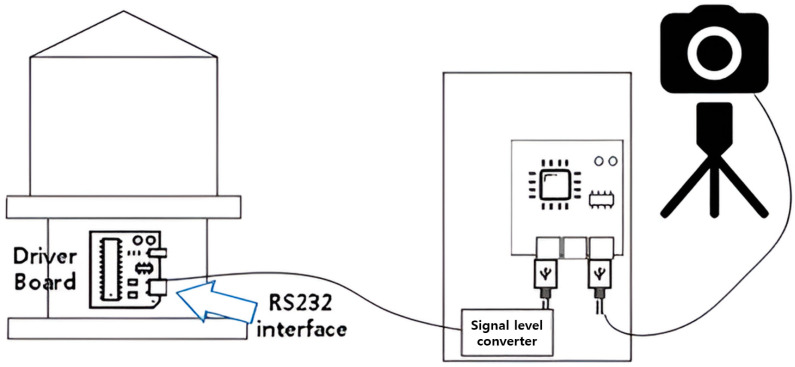
Marine lantern embedded board sea fog video/image-camera linkage system structure diagram.

**Figure 9 micromachines-14-00342-f009:**
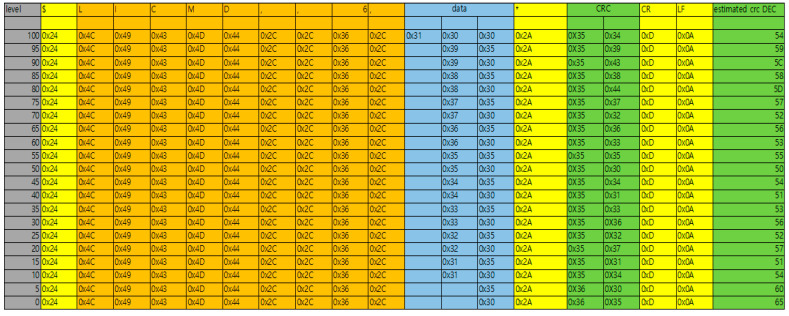
Packet table for light intensity control.

**Figure 10 micromachines-14-00342-f010:**
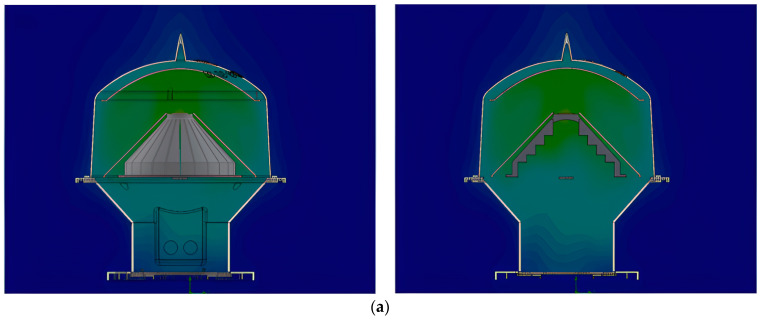
(**a**) LED temperature distribution analysis; (**b**) LED marine lantern used in the practice and modeling results.

**Figure 11 micromachines-14-00342-f011:**
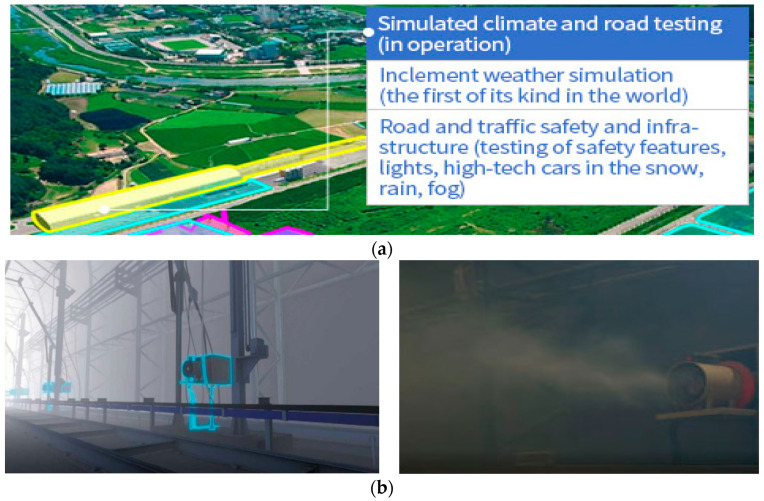
(a) KICTMREC; (**b**) haze generator; (**c**) measurement location.

**Figure 12 micromachines-14-00342-f012:**
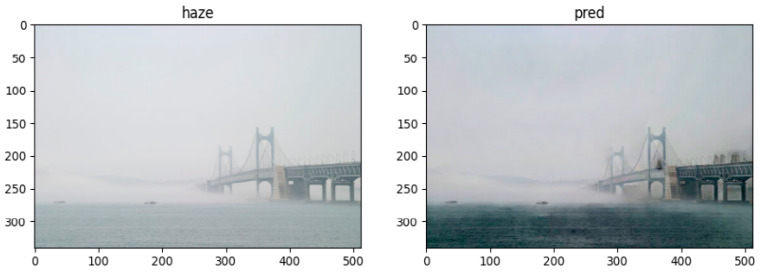
Dehaze algorithm test result.

**Figure 13 micromachines-14-00342-f013:**
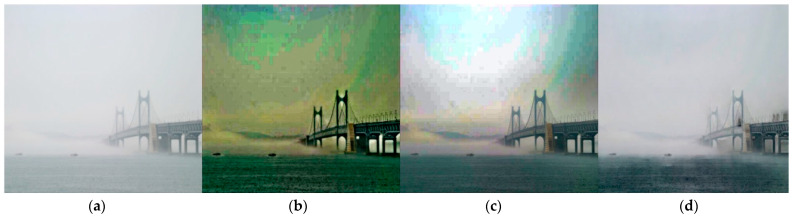
(**a**) original; (**b**) FVR; (**c**) DCP; (**d**) our algorithm visual comparison result.

**Figure 14 micromachines-14-00342-f014:**
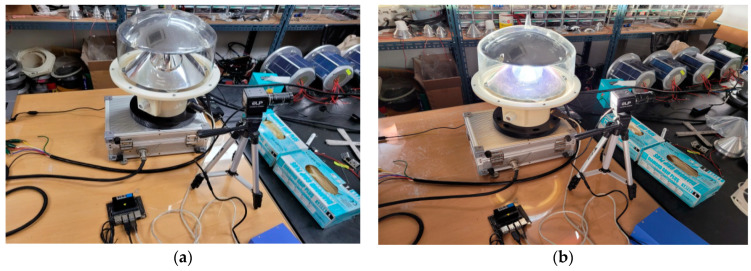
(**a**,**b**) Change in light intensity according to sea fog concentration value input.

**Figure 15 micromachines-14-00342-f015:**
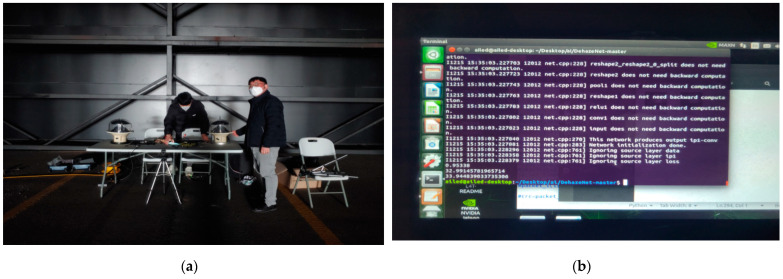
(**a**) Left—intelligent marine lantern; right—conventional marine lantern. (**b**) Results of PSNR and SSIM figures in actual operation.

**Figure 16 micromachines-14-00342-f016:**
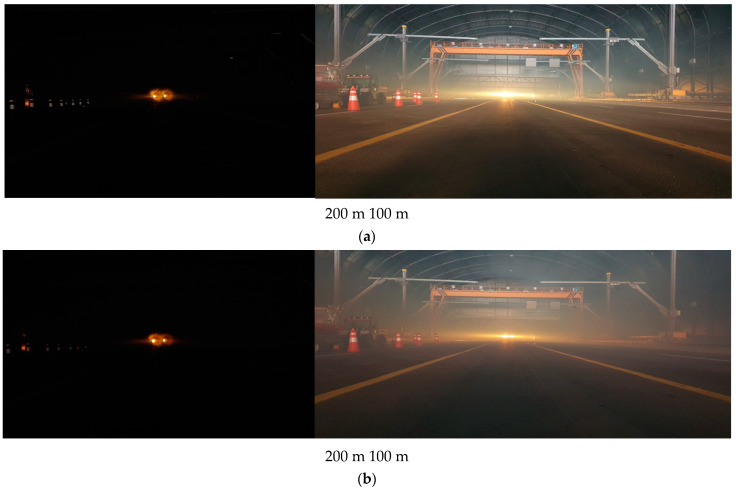
Based on the installed marine lantern, the photo on the left was taken at 200 m and the photo on the right was taken at a distance of 100 m. (**a**) Good visibility; (**b**) 100 m visibility; (**c**) 50 m visibility; (**d**) 10 m visibility.

**Table 1 micromachines-14-00342-t001:** Embedded Board Specification Comparison.

	Jetson Nano	Raspberry Pi 4	LattePanda Alpha
Size	100 mm × 80 mm	85 mm × 56 mm	115 mm × 78 mm
OS	Linux(Ubuntu)	Linux(Ubuntu)	Window 10
CPU	4 Core ARM Cortex A57	4 Core ARM Cortex A72	Intel 8th M3-8100Y
GPU	128 CUDA core (Maxwell)	Broadcom VideoCore IV	Intel HD Graphics 615
RAM	4GB DDR4	1 G~4 GB	8 GB DDR4
Power	10~20 W	15 W	36~45 W

**Table 2 micromachines-14-00342-t002:** Comparison of Dehaze algorithm using sea and lake fog image data sets.

Metrics	DCP	CAP	Dehazenet	AOD-Net	GCAN	AEESFRN
PSNR	18.24	20.78	21.26	22.38	18.16	24.12
SSIM	0.84	0.88	0.87	0.91	0.83	0.93

**Table 3 micromachines-14-00342-t003:** Marine lantern Control Protocol Packet Structure.

Start Packet	Information	Control/Status	Information Delimiter	Marine Lantern ID	Information Delimiter	Control Code	Information Delimiter	Data (N)	Checksum Delimiter	CRC Checksum
‘$’	‘LI’	‘CMD’/ ’STS’/’STE’	‘,’	#1~#256	‘,’	‘1′~’11′	‘,’	Data	‘*’	CRC (ASCII)
1 byte	2 byte	3 byte	1 byte	N byte	1 byte	1 or 2byte	1 byte	N byte	1 byte	2 byte

## Data Availability

Not applicable.

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
