# Peer review of "Enhancement of Marine Lantern’s Visibility under High Haze Using AI Camera and Sensor-Based Control System"

_micromachines, 2023, doi:10.3390/mi14020342_

Round 1
Reviewer 1 Report
In this paper, the authors proposed AI camera and sensor-based control system to enhance the marine lantern’s visibility. The system is composed of a camera sensor, an embedded board, and an LED marine lantern. The removing sea fog from the image was derived using deep-learning-based de-haze technology. Generally, the proposed technique is interesting. The paper is organized well. However, the following problems should be addressed.
(1) In the introduction part, more researches related to the enhancement of the marine lantern’s visibility should be summarized.
(2) What are main innovations or advantages of the Image de-haze technique used in this paper? It sees that the deep learning de-haze technology used in this paper such as e FFA-Net and Dehazenet are the common methods.
(3) The authors states that the PSNR and SSIM are employed to evaluate the sea fog concentration. However, there are not much details analysis in the part of “3. Results”.
(4) The training process of the Dehazenet model used in this research should be detailed described.
(5) I think the framework of the FFA-Net in Fig.6 is unnecessary to be plotted since is was not applied in this research.
(6) Please check the typos of this paper.
Author Response
Hello.
Sorry for the late reply.
Time was delayed because I wanted to give a high-quality answer based on the reviewer's opinion.
Text in red is our response, added to Manuscript.
Please see the attachment.
We look forward to your positive reply.
Happy new year

Reviewer 2 Report
This is an interesting study to apply a deep learning-based dehazing model to sea fog images. A detailed hardware implementation system is proposed.
There are two major concerns regarding this paper.
First, the image dehazing model is the essential component of the system. However, the existing Dehazenet [2] model is used. Please provide justification for this choice and compare it with other models in terms of image quality and hardware implementation. Please discuss how this model is re-trained or tuned for the authors' dataset. Please discuss whether there is any customization or changes using this model [2].
Second, there is no experimental comparison. This paper just "presents" the proposed system without any justification or comparison to evaluate its performance.
In addition, the following are some suggestions regarding the paper's writing.
Figure 2: Please revise the caption inside the figures.
Figure 5: How to obtain the sharpness and brightness values for these photos?
Figures 8, 9, 10 can be removed because these are well-known facts about the PSNR and SSIM.
Author Response

(The authors gave the same response as above.)

Round 2
Reviewer 1 Report
The quality of the revised manuscript is improved. It can be accepted for publication.
Author Response
안녕하세요. 관심을 가져 주셔서 감사합니다. 수정사항이 반영되어 논문의 질이 높아진 점은 긍정적이라고 생각합니다. 첨부 파일을 참조하십시오. 고맙습니다.
Reviewer 2 Report
Thanks for the revision. I would like to follow up with the following two questions:
Q1: Please correct the typo "An end-to..." in Table 2.
Q2: Although there are no other works on the same problem as studied in this paper, more experiments could be conducted to "compare" the proposed approach with various hyperparameter settings, training configurations, etc. It would not be a solid study without such comparisons.
Author Response
Hello. Thank you for your interest. I think it is positive that the quality of the thesis has improved by reflecting the revisions. Please see the attachment. Thank you.

Round 3
Reviewer 2 Report
Thanks for your revision.